# Spectroscopy Techniques for Monitoring the Composting Process: A Review

**Andrés Pérez-San Martín** [1,2,3], **Frutos C. Marhuenda-Egea** [4], **Maria Angeles Bustamante** [5] **and Gustavo Curaqueo** [2,3,*]

1   Doctorado en Ciencias Agropecuarias, Universidad Católica de Temuco, P.O. Box 15-D, Temuco 4813302, Chile; andres.perez.sanmartin@uct.cl
2   Grupo de Investigación en Sustentabilidad Agrícola, Universidad Católica de Temuco, P.O. Box 15-D, Temuco 4813302, Chile
3   Departamento de Ciencias Agropecuarias y Acuícolas & Núcleo de Investigación en Producción Alimentaria, Universidad Católica de Temuco, P.O. Box 15-D, Temuco 4813302, Chile
4   Department of Agrochemistry and Biochemistry, Multidisciplinary for Environmental Studies Ramón Margalef, 03690 Alicante, Spain; frutos@ua.es
5   Centro de Investigación e Innovación Agroalimentaria y Agroambiental (CIAGRO-UMH), Universidad Miguel Hernández, 03312 Alicante, Spain; marian.bustamante@umh.es
*   Correspondence: gcuraqueo@uct.cl

**Abstract:** Composting is described as a sustainable alternative to organic waste reuse from the agricultural and household sectors. The organic matter degradation and stabilization product presents great variability due to the waste composition used. Thus, the use of techniques that allow the monitoring of the decomposition process is necessary to determine the quality of the final product. Therefore, this review critically analyzes and updates the chemical spectroscopy methods described for the evaluation of the organic matter transformation in the composting process to optimize and generate amendments for agricultural use. This review examines spectroscopic techniques such as ultraviolet-visible (UV-Vis), infrared (IR), fluorescence, and 13C NMR reported for the evolution and maturation of carbonate structures in composting. In addition, they are used in the study of indicators for monitoring the degradation of complex structures, such as sugars and proteins, for the formation of precursors that are responsible for the formation of highly stabilized substances such as humic and fulvic compounds. Finally, these parameters may be used to elucidate organic matter degradation and its stabilization process, establish patterns that characterize each stage of composting along with its physicochemical characteristics, and monitor potential phytotoxicity levels.

**Keywords:** compost; UV-Vis spectroscopy; IR spectroscopy; fluorescence spectroscopy; 13C NMR

## 1. Introduction

The worldwide population growth is estimated at around 26% by 2050 [1]. This would imply an increase in food production of almost 70% through agricultural systems [2]. This scenario represents a challenge regarding soil nutrition, the capacity to support the intensification level and ecosystem pressure, and the increased organic wastes generated by the industrial and/or household sector [3]. Therefore, an increase in agricultural production could impact soil degradation due to the application of agrochemicals and intensive management [4]. Thus, from the crop's perspective, the nutrient uptake from the soil system can become a limiting factor [5]. Hence, sustainable alternatives are required that are capable of addressing the recycling of nutrients while minimizing the environmental impact [6].

In this sense, composting is an economic and controlled method for handling and converting organic matter from waste. This is defined as an aerobic biological control process of the decomposition of organic matter in a stable amendment with high fertilization value [7]. This occurs as a natural process that can be enhanced by adding different types of organic or

biological substrates that optimize the development of microbial communities and favor the degradation of organic matter [8]. This generates a product high in bioavailable nutrients, free of pathogenic microorganisms and phytotoxicity with potential agronomic use [9], which is considered a natural alternative to chemical fertilizers, involving the concept of the circular economy [10].

Furthermore, due to the variability and characteristics of the residues used in the composting process, different parameters are analyzed to determine the stability, maturity, and quality of the composted product in each bio-oxidative and maturity stage [11]. This includes various analytical methodologies such as chromatography techniques, spectroscopy, or microbiological sequencing associated with the metabolic response, which allow the evaluation of their physical-chemical and biochemical process [12,13]. However, using a single method to investigate the decomposition process is insufficient. Integrating spectroscopic techniques to complement traditional methods is recommended as a better way to characterize the humification degree and organic matter stability [14]. Thus, spectroscopy methods have been incorporated due to their multiple advantages, such as short analysis time, minimal sample quantity, non-destructive sample, environmentally friendly, and experimental reproducibility [15].

These analyses include UV-Vis spectroscopy to evaluate the organic matter humification degree [16]. Infra-red (IR) spectroscopy for the monitoring of functional groups associated with the transformation of organic substances [17]. Fluorescence spectroscopy for the analysis of fluorescent compounds such as amino acids, proteins, and humic and fulvic substances [18]. Nuclear magnetic resonance (NMR) spectroscopy is associated with the study of carbon structures to analyze the transformation of aliphatic compounds and aromaticity degree [19]. Thus, spectroscopy methods are established as comparative and descriptive methods that allow the complementation of results for monitoring the composting process and its different residues [15].

In turn, compost quality standards are associated with the transformation and humification of organic matter; otherwise, they could generate serious levels of phytotoxicity due to the presence of non-stabilized substances and pathogens [20]. Furthermore, the formation of humic and fulvic compounds is an indicator of highly condensed structures rich in carboxylic, phenolic, hydroxylic, and quinonyl groups of potential benefit in soils and crops [21,22]. However, some studies suggest that traditional humic substance fractions may be assigned to intact and degrading microbial and plant biopolymers [23].

Therefore, this review is focused on a critical and updated analysis of the chemical spectroscopy methods described for the evaluation of the organic matter transformation in the composting process to optimize and generate amendments for agricultural use.

## 2. Composting Process

Composting has become an important alternative worldwide for the reduction and reuse of waste generated in the industrial and household sectors. Highlighted as a sustainable, economical, and controlled method for managing organic matter degradation [24] to increase eco-friendly practices and circular economy within productive processes [25]. It is defined as an aerobic biological process of organic matter decomposition to a stable form with a high content of humic and fulvic substances [21]. A transformation that occurs naturally and can be enhanced or accelerated by adding different organic waste types and inoculums that optimize microbial growth and favor substrate degradation [26,27].

On the other hand, the process effectiveness and product quality will be related to the development of each stage. First, the bio-oxidative stage consists of three phases (mesophilic, thermophilic, and cooling) differentiated by the temperature reached during the decomposition process of simple and complex organic matter [28]. Then, it ends with the maturity stage, where the reorganization and condensation of organic matter into stabilized compounds such as humic and fulvic substances is highlighted [29].

Thus, among the characteristics provided by mature compost to the soil matrix are its physical properties. It highlights that facilitating soil management, generating an

adequate porosity that improves water retention and gas exchange, reduces the risks of erosion, improves the physicochemical characteristics of the soil, enriches the movement of nutrients, helps regulate soil temperature, and reduces water evaporation, which regulates soil moisture by maintaining a stable balance of microorganisms [7].

Furthermore, the benefits of composting on soil chemical characteristics have been reported [30]. This process contributes to improving the bioavailability of nutrients and optimizes their consumption through a more controlled release, avoiding over-fertilization of the crops [31]. It also improves the cation exchange capacity, increases soil fertility, and is a precursor of biostimulant substances for the reduction of the phytotoxic effects of residues caused by high concentrations of salinity, heavy metals, antibiotics, and/or pathogenic microorganisms [32,33].

Besides, compost is beneficial for biological and enzymatic activity, being a suitable medium for the development of microbial communities and as a contribution of microorganisms such as bacteria and fungi to the soil matrix, which also have suppressive properties capable of competing against plant pathogens [34].

Optimal development of these properties requires monitoring some key parameters associated with maturity and stability parameters in the composting process (Figure 1). Factors such as pH and temperature are the most important characteristics during composting [26]. These parameters define the metabolic processes and the succession of microbial communities at each stage [35]. Initially, vegetable or food waste is characterized by lowering the pH in the pile due to the release of organic acids in the early stages of degradation [36]. Also, incorrect storage of these wastes can cause the accumulation of acidic substances due to fermentation processes. Consequently, low pH levels in the early stages of composting can inhibit the degradation of organic matter [37] and decrease the temperature of the pile, delaying the transition to the thermophilic stage.

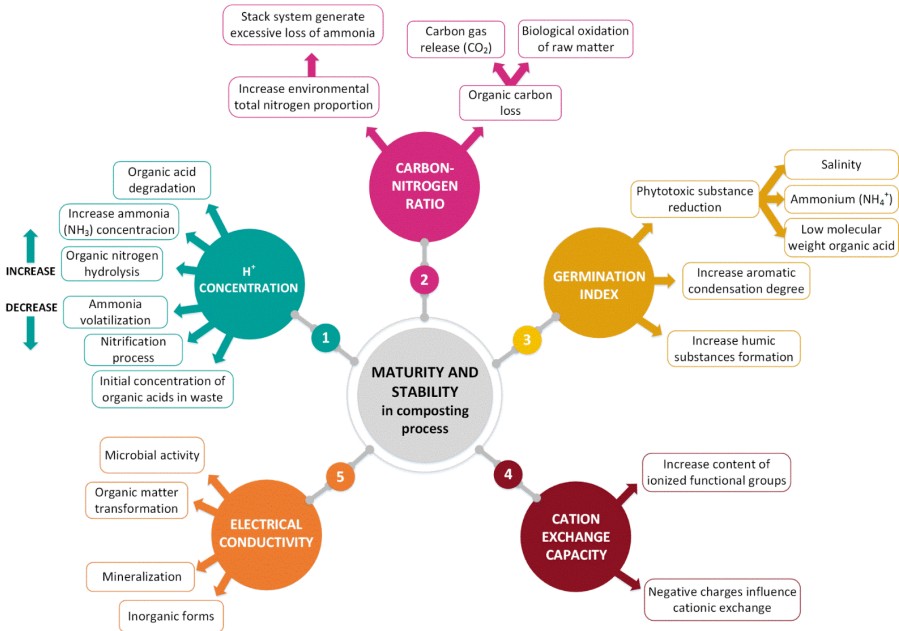

**Figure 1.** General scheme of physicochemical parameters associated with the composting process.

In turn, organic matter is a key factor to consider during the composting process, a parameter related to the C/N ratio that controls the evolution of the microbial community and humification processes [5]. Optimal initial C/N values range from 20 to 30, which avoids N mineralization and immobilization processes [38]. Consequently, the decrease in organic matter content, $CO_2$ formation, N ammonification processes, and volatilization as $NH_3$ form generate a variation in the C/N ratio during the composting process [29].

On the other hand, electrical conductivity reflects the waste mineralization process, and this is associated with salt formation due to the organic matter transformation into

inorganic matter by the action of microorganisms [39]. Moreover, cation exchange capacity increases due to the content of polar functional groups such as hydroxyl, carboxyl, and methoxy by the action of raw material biological oxidation [40]. These parameters have a final effect on the germination rate, affecting the stability and maturity of the compost produced, decreasing the concentration of the phytotoxic compound, and favoring the formation of the humic substance [41].

Furthermore, Figure 2 summarizes the main products obtained by the degradation of microbial communities at each stage of the composting process. These results have been adapted from [21,42] in which different ways of formation of humic compounds are proposed considering the environmental variations. The synthesis of these substances is characterized by their particular resistance to degradation and their beneficial properties of agronomic interest in soil quality and against environmental phytotoxicity [21]. Its formation is an indicator parameter of compost stability and maturity, but it has not yet been completely elucidated due to the complexity of the associated biochemical processes [43]. For this reason, some theories have been described in this respect, such as the depolymerization and polymerization theories called lignin-protein theory and sugar-amine condensation theories (Maillard), respectively [44]. The polyphenol theory [21] has been widely accepted, leading to the combination of assumptions and the lignin/phenol-protein theory [45].

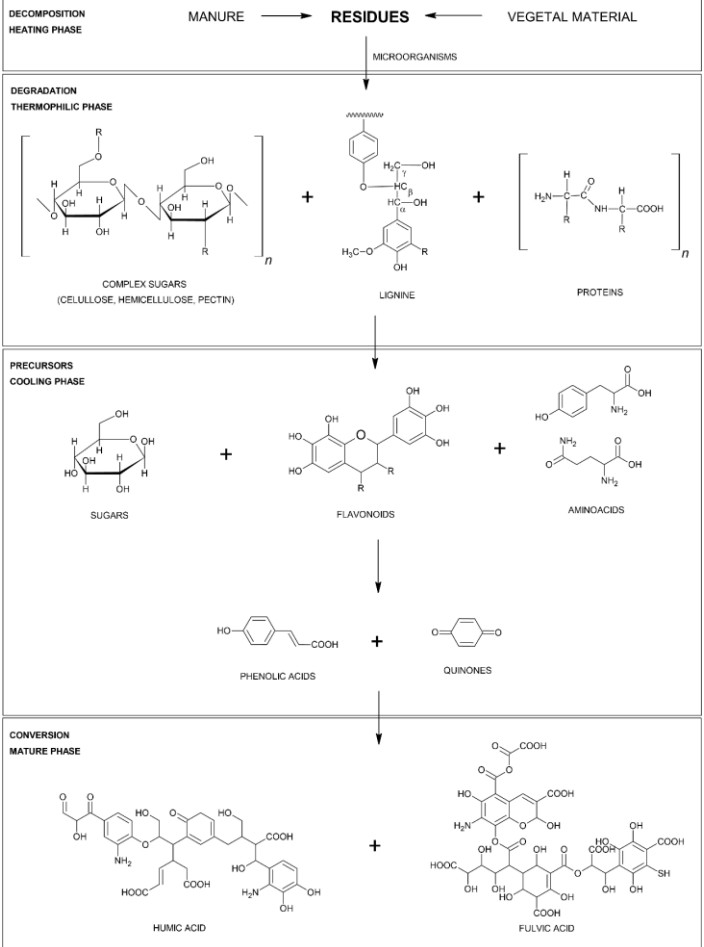

**Figure 2.** Transformation mechanism of organic matter in each stage of the composting process [21,42].

## 3. Ultraviolet-Visible Spectroscopy (UV-Vis)

UV-Vis spectroscopy in composting allows for the detection of the variation in the absorption bands during the process. These bands are related to the degradation of organic matter. They are interpreted as the change of absorption at a given wavelength

by the effect of the variation of the electronic structures of the compounds [46]. The UV-Vis spectra have different absorption regions: 280 nm indicates the beginning of the transformation of aliphatic compounds and lignins, 465 nm indicates the beginning of the humification process by depolymerization of complex organic molecules, and 665 nm indicates process humification with high oxygen content and aromatic groups [47,48]. In the UV absorption ranges, high absorbance at wavelengths less than 250 nm is characteristic of humic substances. It is attributed to the absorption of radiation by the double bonds present in aromatic C=C and ketone C=O [49]. For a wavelength of 280 nm, total aromaticity is detected due to the electron transition occurring in the 270–280 nm region for phenolic arenes, benzoic acids, aniline derivatives, polyenes, and polycyclic aromatic hydrocarbons with two or more rings [50,51]. Moreover, the absorption band at 365 nm is estimated to be related to the presence of fluorescent humic and fulvic compounds [28].

These parameters have also been interpreted as the relationship ratio between the regions described (Table 1). The first one indicates the ratio between the beginning of the organic matter transformation process and the humification degree. It is expected that the latter will increase at the end of the process, and the transformation process will decrease over time [52]. Thus, research reports the relationship E250/E203 as a measure to evaluate the substitution of aromatic rings and the presence of aliphatic groups. Therefore, polar functional groups are found at higher rates in aromatic rings, such as hydroxyl, carbonyl, and carboxyl groups [53,54]. In turn, the absorbance ratio of E250/E365 has been described as a way to characterize the aromaticity and the polycondensation degree of organic molecules [55]. The E280/E665 index has been reported to indicate the HA high molecular weight poly-condensed aromatic nucleus [43]. Thus, the E465/E665 index is related to the depolymerization process of organic matter and its transformation into stabilized compounds [52]. In this sense, its decrease is attributed to the absorption of C=C aromatic functional groups, and its increase suggests a low molecular weight and chromophores presence as carboxylic, ketonic, and aromatic groups process [56].

Moreover, E250/E365 and E465/E665 indexes are inversely related to the degree of aromaticity and molecular weight of soil humic substances, i.e., related to that low ratio values are characteristic of humic substances fraction and high humification degree [48]. Furthermore, E250/E436 values are associated with the presence of humic substances derived from organic matter of terrestrial (allochthonous) or microbially derived (autochthonous) origin [51]. Therefore, these indices are related to an increase in the polycondensed aromatic nucleus of humic acids, and high values may suggest the compost formation with high concentrations of phenolic groups, benzene-carboxyl, and methoxy in its structure [43,57].

Some authors [48,51] describe that the relations between E250/E365 and E465/E665 present an inverse tendency with aromaticity, acidity, molecular weight, and proportion in the content of oxygen, carbon, and carboxyl groups. Low values indicate a high condensation degree of aromatic humic components, high molecular weight, and low acidity, characteristic of the humic acid fraction. On the contrary, a high ratio indicates the presence of aliphatic compounds, low molecular weight, and high acidity, characteristic of the fulvic acid fraction.

In turn, another parameter has been described to evaluate organic matter transformation. Specific ultraviolet absorbance (SUVA) is defined as absorbance at a given wavelength normalized by dissolved organic carbon concentration (DOC) [58]. Aromaticity degree can be associated with ranges between 250–280 nm, and $SUVA_{254}$ is directly related to the percentage of aromaticity of humic substances and molecular weight [59]. Furthermore, $SUVA_{280}$ has been used to describe the number of aromatic compounds at each stage [60].

**Table 1.** Summary of the UV-Vis absorption regions described for the composting process.

| Indicator | Description | References |
|---|---|---|
| **Absorption region (nm)** | | |
| 250 | Absorption of aromatic C=C and ketone groups (C=O). | [49] |
| 280 | Evaluate the beginning of the transformation of aliphatic compounds and lignin—absorption of aromatic groups. | [51] |
| 365 | Related to fluorescent humic-like and fulvic-like substances. | [28] |
| 465 | Indicates the beginning of the process of humification and depolymerization of complex molecules. | [47,48] |
| 665 | Indicates process humification, oxygen content, and aromatic groups. | |
| $SUVA_{254}$ | Related to the percentage of aromaticity of humic substances and their molecular weight. | [59] |
| $SUVA_{280}$ | Describes the amount of aromaticity present at the stages evaluated. | [60] |
| **Ratios** | | |
| E250/E203 | Substitution measure of aromatic rings. The high degree indicates a greater presence of polar groups. | [53,54] |
| E250/E365 | Evaluates the degree of aromaticity of organic molecules and inverse to the degree of humification. | [55] |
| E250/E436 | Associated with terrestrial origin of humic substances (allochthonous or autochthonous). | [51] |
| E280/E665 | Indicates the transformation process of humic substances, maintaining a decreasing trend over time. | [43,52] |
| E465/E665 | Related to the degree of condensation and aromaticity of humic compounds. | |

## 4. Infrared Spectroscopy (IR)

IR spectroscopy is obtained from the molecular vibration generated by the different functional groups [61]. In the characterization of the composting process, peaks have been described that can contribute to monitoring the conversion stages of organic matter.

Table 2 describes the main functional groups attributed to these bands and their vibrations. The bands 2964 to 2930 cm$^{-1}$ describe aliphatic carbon groups attributed to the presence of fatty acids [62]. The constant presence of this signal may be due to the concentration of biodegradable plant material composed of lignins, cutins, and suberins [63].

The signal between 1590 to 1504 cm$^{-1}$ corresponds to the presence of amide II groups and the lignin and cellulose present in residues containing plant materials such as plants or wood, which are rich in this compound [64]. Another typical band of lignin is caused by the vibration of the aromatic (C=C) skeleton [65]. On the other hand, a band of protein origin can be found in nitrogen-rich composts associated with amide II groups [66]. Overall, the decrease of the 2964 and 1571 cm$^{-1}$ bands may indicate rapid biodegradation of aliphatic and amide groups in the composting process, and the peak of 1408 cm$^{-1}$ demonstrates the degradation of organic carbon belonging to carboxyl groups [67,68].

Besides, it is possible to generate a relationship between these bands using the relative intensities of each peak. Relationships between 1642/1111 and 1642/1003 ratios indicate the presence of (aromatic C/alcohol C) and (aromatic C/polysaccharide C), respectively. The increase in these values demonstrates that biodegradation of polysaccharides and groups such as alcohol and ether occurs [53]. The transformation of aliphatic carbons to their stabilized form as aromatic carbons due to the degradation process is related to the increase in the 1642/2964 (aromatic C/aliphatic C) ratio [57]. The increase in these ratios is associated with the degradation of organic matter as an energy source for microorganisms, generating an increase in aromatic products during the composting process [28].

The 1387/1544 (phenolic C/aromatic C-N) ratio suggests a direct evaluation of humification degree and the state of maturity due to the increase in aromatic carbon associated with OH groups (phenols), carboxylates (COO$^-$), quinones, amides, and nitrates [52]. Finally, the C/N ratio can be attributed to the change in the bands of the 1037/1384 ratio due to carbon source degradation and medium alkalization by the ammonia groups effect [66].

**Table 2.** Summary wavenumber described for the evaluation of the composting process.

| Wavenumber (cm$^{-1}$) | Description | Reference |
|---|---|---|
| 3437–3263 | This occurs due to stretching vibration produced by OH groups of alcohols, phenols, and organic acids. | |
| 2964–2930 | Bands corresponding to C-H stretching and asymmetric vibrations in aliphatic structures. | |
| 1652–1642 | Bands produced by stretching vibrations of C=C bonds in aromatic structures by ketone groups such as quinones and amide groups (C-N). | |
| 1590–1500 | Obtained due to the deformation and vibration stretching of amide II groups (N-H) and (C-N) of secondary amides, respectively. | [28,52,53,57,66] |
| 1408 | Intensity assigned to the vibration asymmetric stretching of carboxyl groups (C-O). | |
| 1387 | Described by the deformation of phenolic OH groups and aromatic alcohols, by the asymmetric stretching of carboxyl ions (COO$^-$) of disubstituted aromatic rings, and by the presence of inorganic nitrogen as nitrates. | |
| 1116–1003 | Characteristics of C-O bond stretching vibrations in structures such as polysaccharides, ethers, and secondary alcohols. | |

## 5. Fluorescence Spectroscopy

Fluorescence spectroscopy is a nondestructive method for characterizing organic matter in relation to its important fluorescent compounds, which include humic-like, fulvic-like, and proteinaceous substances [69]. This method is based on the signal interpretation by light emission effect in the process of electronic de-excitation of conjugated systems [70]. It is used to evaluate the humification degree of organic matter in the composting process. Fluorescence in organic matter is related to the condensed aromatic rings presence of unsaturated aliphatic carbon chains [71]. Fluorescence bands at long wavelengths have been described by high molecular weight and complex structural compounds with a high conjugation degree, and fluorescence bands at shorter wavelengths have been described for simple structures and small degrees of conjugated chromophores due to a low degree of aromatic polycondensation [72]. Therefore, the stability of organic matter and the increase in its condensation degree is associated with a chemically stable structure with greater residence in the environment, which contributes to enhancing the structure and fertility of the soil [73]. Thus, different parameters can be obtained from fluorescence spectroscopy associated with organic matter transformation in each stage of the composting process: Emission spectra, Excitation spectra, Synchronous-scan excitation spectra, and Excitation-Emission matrix (EEM).

### 5.1. Emission and Excitation Spectra

Emission spectra are generated due to the sample's exposure to a specific excitation wavelength and are represented by one or more emission bands. The excitation wavelength for obtaining emission spectra in the composting process analysis is 254 nm. The results reported a fluorescence peak at 340 nm in the initial phase, changing to 430 nm during the process, indicating an increase in the aromatic group condensation due to the maturation of the compost [55]. Thus, the relationship between emission spectra at 450 and 500 nm with an excitation wavelength of 370 nm ($F_{450}/F_{500}$) has been described. This index investigates the organic matter sources [74] and correlates with humic substances origin. A value less than 1.4 indicates terrestrially derived humic substances, and larger than 1.90 indicates microbially derived humic substances [75]. A high value of this ratio indicates weak aromatic humic substances with less benzene ring structure [76]. In this sense, monitoring of humic substances evaluation has been reported at an excitation wavelength of 360 and 470 nm and fulvic substances at an excitation wavelength of 360 and 468 nm [54,77].

Contrary to emission spectra, excitation spectra are determined using a fixed emission wavelength and the excitation scan at different wavelengths. Moreover, the results obtained can be assimilated to the UV-Vis absorption spectra, the information from which can be used to generate the emission spectra [78]. Studies [14] describe a composting of municipal solid waste, where they evaluate an excitation spectrum of 300 to 500 nm using an emission wavelength of 520 nm. The results reported four major peaks, of which only two were representative. One of these, around 436 nm, was attributed to aromatic rings bearing electron donor groups, and the second at 383 nm, possibly due to fluorophores originating from the polycondensation of carbonyl groups and lignin-derived phenolic structures [79]. These two peaks were associated through the fluorescence ratio determination ($I_{436}/I_{383}$) that showed an inverse correlation to the SUVA$_{254}$ indicator and the emission at 351 nm, which are related to the abundance of aromatic carbons and the organic matter humification, respectively [80,81]. In addition to the evaluation of compost, excitation spectra for humic and fulvic acid fractions have also been described. Both describe an excitation range of 300–800 nm, differentiated by the emission wavelength at 590 and 530 nm, respectively [77]. Therefore, it is suggested that these parameters can be used to indicate the degree of stabilization of organic matter associated with its humification degree in the composting process.

### 5.2. Synchronous-Scan Spectra

Synchronous-scan spectra are generated from the simultaneous detection of excitation and emission scans at a predetermined wavelength difference, favoring signal amplification in trace compounds [82]. In turn, compared with other fluorescence analyses, it is possible to obtain a unique and clear spectrum associated with specific functional groups and chemical structures [83]. Synchronous fluorescence represents the spectra summation of different fluorophores in dissolved organic matter and exhibits better resolution than excitation and emission spectra [14]. This analysis reports the presence of characteristic peaks (Table 3) for the composting process associated with organic matter biodegradation and biosynthesis of humic-like substances [28]. First, a high-intensity Peak A has been reported in the range of 250–308 nm related to the presence of protein-like substances and mono-aromatic compounds, which tends to decrease in the process [28,72].

Furthermore, a Peak B was reported between 308–365 nm attributed to the presence of fulvic-like indicates the presence of polycyclic aromatics with three to four fused benzene rings and two to three conjugated systems in unsaturated aliphatic structures [84–86]. The identification of humic-like substances has been reported in Peak C 363–595 nm [14,28] related to the presence of polycyclic aromatic compounds with approximately 5–7 fused benzene rings tend to increase in intensity as the compost matures [85,87]. In turn, fluorescence techniques have been described for the characterization of humic substances in soil analysis due to their strong selectivity, high sensitivity, and no damage to the sample [88].

**Table 3.** Summary synchronous-scan excitation regions described in the composting process.

| Region (nm) | Description | Reference |
|---|---|---|
| 250–308 | Describes the presence of protein-like substances and mono-aromatic compounds, which tend to decrease during composting. | [28,72] |
| 308–365 | Attributed to the presence of fulvic-like acid and polycyclic aromatics with fused benzene rings and conjugated systems in unsaturated aliphatic structures. | [84,86] |
| 363–595 | This region is associated with polycyclic aromatic compounds with fused benzene rings, which increase in proportion to the maturation of the compost. | [14,28] |

## 5.3. Excitation-Emission Matrix

Excitation–emission matrix (EEM) spectra are obtained by subsequently scanning the emission spectra range by increasing the excitation wavelength. Figure 3 compiles the Excitation-Emission matrix results reported by [89]. These results are represented in a graph divided into five regions. The dissolved organic matter of each region represents particular structures, in the case of the composting process, which can increase or decrease the intensity of fluorescence as it moves towards its maturity stage [68]. These peaks have been mainly associated with organic compounds such as humic and fulvic acid-like amino acids, proteins, or phenolic structures [90].

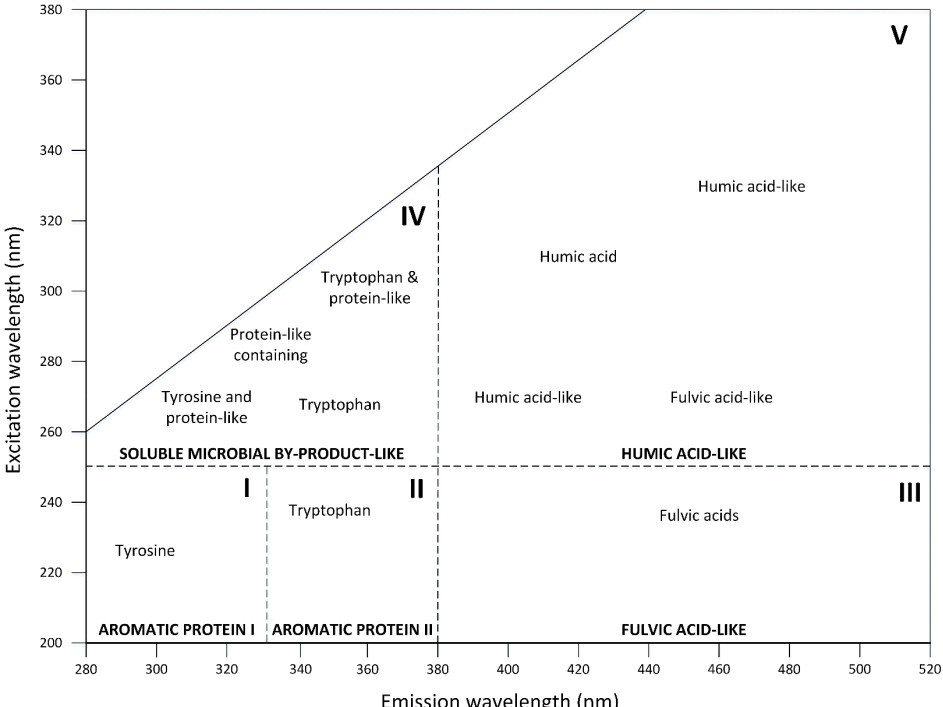

**Figure 3.** Location of structures for each region in an Excitation-Emission Matrix (EEM) of organic matter (adapted from [89]).

In this sense, Regions I and II are related to amino acids derived from simple aromatic proteins such as tyrosine and tryptophan with peaks at shorter excitation-emission wavelengths (<250 nm/<350 nm); Region III is described by peaks at shorter excitation wavelengths and longer emission wavelengths (<250 nm/>380 nm) are related to fulvic substances; Region IV is associated with microbial by-product-like material as a result of microbial protein degradation, reporting peaks at intermediate excitation, and shorter emission wavelengths (250~280 nm/<380 nm); Region V includes a longer excitation range and longer emission wavelengths (>280 nm/>380 nm) and is directly associated with humic substances and may even contain part of fulvic substances fluorescence, depending on the intensity of these [14,68,91,92].

In the first stage of composting, it is possible to observe a peak in wavelength pair 219/348 nm (Ex/Em) of greater intensity that extends to Region I, which disappears early in the process and does not report results in the maturation stage [93]. In Region III, a peak in an Ex/Em wavelength pair 220/429 nm can emerge in the early stages of composting and is a monitoring point for fulvic substance formation, reaching a maximum peak during the maturity stage [14]. Then, the quantity of fulvic acids could decrease due to their transformation into humic substances [94]. On the other hand, Region IV registers at the beginning of the composting process a weak signal corresponding to an Ex/Em wavelength pair 278/354 nm that progressively disappears until the maturation stage due to the effect of microbial activity on the organic matter degradation [95]. Finally, in Region V, a peak in

Ex/Em wavelength pairs 313/406 nm can be observed, which increases significantly in the early stages of composting with an increase in intensity in the maturity stage, associated with a high degree of process humification [96]. Furthermore, it is possible to observe a peak at an Ex/Em wavelength pair 289/421 nm, indicating the maturation stage due to the organic matter stabilization, which does not occur in earlier stages [14,80,81].

## 6. Nuclear Magnetic Resonance (13C NMR)

Nuclear magnetic resonance (NMR) is produced by the reorganization of nuclear spin due to the application of a magnetic field to the nuclei at the low-energy level, which will absorb electromagnetic energy and transit to the high-energy level. When the electromagnetic field disappears, the nucleus releases the absorbed energy, returning to its low-energy state in a process called relaxation. This energy is detected as radiofrequency detects the presence of carbon and its relationship with the functional groups [97].

During the composting process, chemical shifts have been described to monitor the organic matter decomposition and to evaluate structures stabilized over time. Figure 4 summarizes the main chemical shifts reported in the literature, corresponding to carbon monitoring as Alkyl-C (45-0 ppm), methoxyl C (60–45 ppm), aliphatic C substituted with O or N (110–60 ppm), aromatic C (110–145 ppm), phenolic C and aromatic ether substituted with O or N (160–145 ppm) and carbonyl C, carboxyl C, quinones, ester, and amides carbon (210–160 ppm) [19,52,98,99].

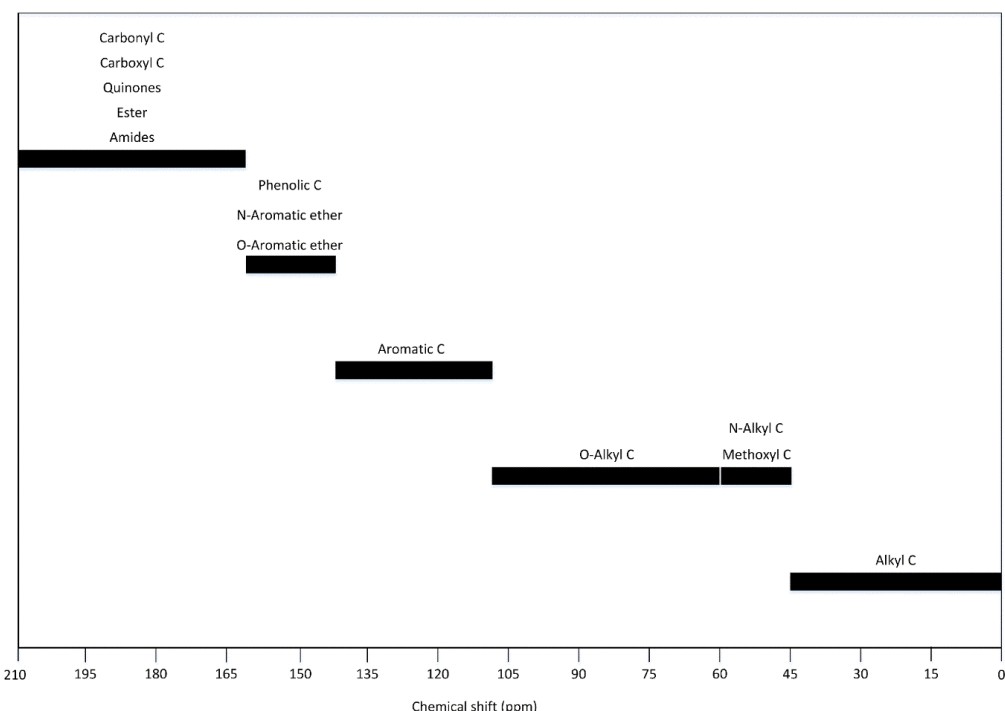

**Figure 4.** Signal distribution 13C NMR about the chemical shift described in the composting process [19,52].

Studies performed by [19] describe an increase in polar carbonate structures, such as carboxyl, carbonyl, O-alkyl C, aromatic C, and phenolic C groups, corresponding to the formation of humic and fulvic substances. In contrast, Alkyl C shows a decreasing trend due to the degradation of carbonate structures in the preliminary stages, which can be ascribed to the aliphatic carbons of the methyl terminal groups, lipids, fatty acids groups, and other biopolymers [100]. Besides, O-alkyl C may correspond to carbohydrate carbons substituted with oxygen atoms and polysaccharide structures such as aliphatic alcohols or cellulose [101]. In the case of methoxyl groups and aromatic ethers, it has been reported that they may not present significant changes in the composting process because they are

structures resistant to microbial degradation. However, they are structures that participate in the formation of the humic acid [102].

From these chemical shifts, it is possible to determine indicators such as the aromaticity index (Equation (1)), which is reported as the relationship between aromatic C groups and alkyl C groups with a tendency to increase in the final stages of composting [98,103].

$$\text{Aromatic index} = \frac{(\text{Aromatic C} + \text{Phenolic C})}{(\text{Alkyl C} + \text{O alkyl C} + \text{N alkyl C} + \text{Aromatic C} + \text{Phenolic C})} \times 100 \tag{1}$$

In addition to this indicator, the lignin ratio (Equation (2)) is described as a relationship between methoxyl C and phenolic C to evaluate the degradation of complex carbon structures in the composting process, demonstrating a decrease over time due to the increase of phenolic C structures [99].

$$\text{Lignin ratio} = \frac{\text{Methoxyl C}}{\text{Phenolic C}} \tag{2}$$

Thus, among the most important aspects to evaluate the maturation degree of the compost is the germination rate, which is one of the most important parameters to determine the stability of the degradation of the organic matter [95]. In this sense, the Humification Index (HI = Fulvic acid carbon/Total organic carbon) and the degree of polymerization (DP = Humic acid carbon/Fulvic acid carbon) are directly related to the germination rate, while HI and DP are positively correlated with the chemical shift aromatic C, phenolic C and O-alkyl C and negatively with C/N ratio, the latter positively correlated with alkyl C [19]. Therefore, it is estimated that aromatic C, phenolic C, and O-alkyl C could be the most important functional groups for the evaluation of the humification degree, and a lower C/N ratio and alkyl C intensity must be obtained at the stage of maturation [103,104].

## 7. Future Perspectives

Monitoring the composting process evolution is a key factor in evaluating the degree of stability and maturation. These parameters change depending on the conditions of the process (oxygenation, humidity, temperature) and the characteristics of the material or residues used. For these reasons, the high variability that can present a product of another requires the standardization of these conditions to reach the quality points established in each compost. Thus, spectroscopy techniques (Table 4) are an alternative to monitoring the relationship between compost maturity state and synthesis of highly stabilized substances with potential agronomic interest.

**Table 4.** Summary of the main characteristics of the spectroscopy techniques reviewed.

| Spectroscopy | Description | Advantages | Disadvantage |
|---|---|---|---|
| UV-Vis | An electronic transition effect occurs due to the absorption of energy by the electrons (transmittance). | - Cost-effective<br>- Easy method<br>- Simple to interpret | - Liquid sample only<br>- Preliminary qualitative analysis<br>- Absorption peaks superposition |
| IR | The energy absorbed by the functional groups produces different types of molecular vibrations (reflectance). | - Determines liquid and solid samples<br>- Conclusive results | - Band overlap<br>- Interpretation may be difficult<br>- Solid sample recommended |
| Fluorescence | Based on the signal interpretation by the effect of light emission in the process of electronic de-excitation of conjugated systems (emission). | - Different analyses are available<br>- Allows compounds tracking<br>- It allows humic substance analysis | - Previous knowledge is required<br>- More complex interpretation results<br>- Liquid sample recommended |
| NMR | Produced by the reorganization of nuclear spin due to the application of a magnetic field to the nuclei (emission). | - Non-Destructive<br>- Minimum sample treatment<br>- Complementary analysis | - Expensive<br>- Signals superposition<br>- Solid sample |

Moreover, maintain monitoring residues with potential environmental risk effects such as sludge or sewage material. These substances may contain a high concentration of heavy metals, antibiotics, or the presence of microorganisms pathogenic to humans.

Therefore, it requires monitoring to evaluate the reduction of factors that may subsequently affect crops and human consumption, integrating spectroscopy analysis in situations that make it difficult to determine these substances.

Finally, few studies have reported on the use of Raman spectroscopy in compost. Ref. [105] discusses this absence from the perspective of soil analysis. This technique allows a complimentary analysis of IR spectroscopy, identifying the transitions between the state of rotation energy and molecular vibration and being able to characterize functional groups in organic and inorganic samples. In turn, it focused on the determination of microplastics in agro-industrial wastes and their evolution during the composting process.

Among the advantages of this technique are highlighted: presents a low overlap of bands, allowing better detection and characterization of components; does not require prior sample preparation and has no interference from glass containers or aqueous solutions; also, it is possible to quantify the bands obtained and analyze in situ [106]. Thus, it is estimated that the integration of this technique in future analyses could significantly contribute to the characterization of the composting process and the compounds generated by the degradation of organic matter.

## 8. Conclusions

In conclusion, the composting process has been established as a sustainable alternative within the circular economy for the reduction and treatment of organic waste. Also, it is a useful method for generating soil amendments capable of recycling nutrients from organic waste generated by the agricultural and household sectors. The stabilization of organic matter and correct degradation of complex structures allows the generation of an optimal product, minimizing the probability of environmental impact due to the persistence of phytopathogenic agents or the presence of toxic substances in concentrations that affect soil quality. Spectroscopy techniques such as UV-Vis, IR, Fluorescence, and 13C NMR allow monitoring of the different stages of the composting, showing the stabilization and degradation of organic matter in highly condensed compounds. Each technique establishes a relationship concerning the functional groups and elucidates the part of the structures that are formed as a result of the action of microorganisms on organic waste. Therefore, it is considered that these methodologies establish reliable and complementary quantitative indicators to establish the necessary quality criteria required by a compost.

Finally, more information is needed on studies of Raman spectroscopy in the composting process, as well as a better elucidation of action mechanisms and metabolic pathways associated with each composting process for the formation of humic and fulvic substances. Also, to establish indices to evaluate potential risk substances that affect the germination or contamination of soils and prevent their use as an agronomic amendment.

**Author Contributions:** Writing—original draft preparation, A.P.-S.M. and G.C.; review and editing, F.C.M.-E. and M.A.B.; investigation, A.P.-S.M. All authors have read and agreed to the published version of the manuscript.

**Funding:** This work was supported by the National Agency for Research and Development (ANID)/Scholarship Program/Doctorado Nacional/2020—21201805 and VIPUCT 2020REGGC07 Project by Universidad Católica de Temuco.

**Data Availability Statement:** Not applicable.

**Acknowledgments:** Doctoral Program in Agricultural Sciences of the Universidad Católica de Temuco.

**Conflicts of Interest:** The authors declare no conflict of interest.

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
