# Peer review of "Spectroscopy Techniques for Monitoring the Composting Process: A Review"

_agronomy, doi:10.3390/agronomy13092245_

Round 1
Reviewer 1 Report
This review critically analyzes and updates the chemical spectroscopy methods described for the evaluation of the organic matter transformation in the composting process to optimization and generation of amendments for agricultural use. Especially, examines the use of UV-Vis, IR, fluorescence, and 13C NMR reported for the evolution and maturation of carbonate structures in composting. Overall, the manuscript is well written. But the manuscript still needs some improvement.
1. Please further clarify the relationship between peak/wavelength and compost maturity or stability.
Such as, Line 338-339, Page 9,
“a peak in an Ex/Em wavelength pair 220/429 nm which increases in the maturity stage and decrease due to its transformation into humic substances” does this peak increase or decrease indicate maturity?
2. Please provide the types of samples required for each spectroscopy indicator, solid, liquid or based on DOC. Currently, some were provided, some were not.
3. Please check the occasional disjointed sentences/syntax errors which need to be fixed in this manuscript. Such as:
13C or 13C NMR, Lines 288-291, page 8, …….
English is generally good
Author Response
Thank you for the comments and suggestions.
|
1. Please further clarify the relationship between peak/wavelength and compost maturity or stability. Such as, Line 338-339, Page 9, “a peak in an Ex/Em wavelength pair 220/429 nm which increases in the maturity stage and decrease due to its transformation into humic substances” does this peak increase or decrease indicate maturity? R: We appreciate the comment. This issue was incorporated, from L347 to L351 rewriting the paragraph for more clarity. 2. Please provide the types of samples required for each spectroscopy indicator, solid, liquid or based on DOC. Currently, some were provided, some were not. R: Thank you for your comment. We incorporated the types of samples in all each technique (Table 4) page 12.
13C or 13C NMR, Lines 288-291, page 8, R: We appreciate the comment. All issues were corrected in the manuscript.
|
Reviewer 2 Report
This study revises bibliography on the monitoring of composting process by UV-Vis, IR, fluorometry and NMR. The text is well written and English is correct. Citations are appropriate and actualized. Figures and Tables are also appropriate.
Specific comments
- Line 75: Along the study, authors and cited studies refer to humic substances and humification in several occasions. As operationally units of the soil most recalcitrant organic matter, humic substances has been thought as novel categories of cross-linked macromolecular structures. However, some studies suggest that traditional humic substances fractions may be assigned to intact and degrading microbial and plant biopolymer (Kelleher & Simpson 2006). I suggest adding comments on this.
Kelleher BP & Simpson AJ. 2006. Humic Substances in Soils: Are They Really Chemically Distinct?
Environ. Sci. Technol. 40: 4605–4611.
- Lines 415-417: I agree with suggested future perspectives, except for the sentence: “Therefore, future research could focus on determining the compounds produced during composting maturity stage as structural models of hu-mic substances (humines, humic and fulvic acids) from the use of spectroscopy and chromatography techniques.”
I do not think that the elaboration of structural models of humic substances is a straightforward future perspective derived from your study, that was so focus on a simple spectrophotometric way of monitoring composting process for guaranteeing maturity and stability. I would eliminate this sentence.
- Line 555-556: Citations 25 and 34 are the same: “Cáceres, R.; MaliÅ„ska, K.; Marfà, O. Nitrification within Composting: A Review. Waste Manag. 2018, 72, 119–137, 531 doi:10.1016/j.wasman.2017.10.049.” Correct this.
Author Response
Line 75: Along the study, authors and cited studies refer to humic substances and humification in several occasions. As operationally units of the soil most recalcitrant organic matter, humic substances has been thought as novel categories of cross-linked macromolecular structures. However, some studies suggest that traditional humic substances fractions may be assigned to intact and degrading microbial and plant biopolymer (Kelleher & Simpson 2006). I suggest adding comments on this.
R: Thank you for your comment. We included your suggestion in L78
|
Lines 415-417: I agree with suggested future perspectives, except for the sentence: “Therefore, future research could focus on determining the compounds produced during composting maturity stage as structural models of humic substances (humines, humic and fulvic acids) from the use of spectroscopy and chromatography techniques.” |
I do not think that the elaboration of structural models of humic substances is a straightforward future perspective derived from your study, that was so focus on a simple spectrophotometric way of monitoring composting process for guaranteeing maturity and stability. I would eliminate this sentence.
R: We appreciate your comment. In this sense, we eliminated this paragraph
Line 555-556: Citations 25 and 34 are the same: “Cáceres, R.; MaliÅ„ska, K.; Marfà, O. Nitrification within Composting: A Review. Waste Manag. 2018, 72, 119–137, 531 doi:10.1016/j.wasman.2017.10.049.” Correct this.
R: Thank you for the comment. The mistakes were corrected. In the new manuscript, we assigned citation N° 26 to Cáceres et al., 2018.
Reviewer 3 Report
English editing is required
More in-depth review should be carried out on Synchronous-scan excitation spectra, the information presented is too short.
Also, Fluorescence spectroscopy section should also contain more examples and information
The manuscript needs minimal English editing
Author Response
English editing is required
R: Thank you for the question and comments. The new version of the manuscript was reviewed and corrected by a native English speaker.
|
Also, Fluorescence spectroscopy section should also contain more examples and information. R: Thank you for your comment. We included more information in L251 and new citation. |